

# The impact of changes in LH levels in flexible antagonist protocols on the clinical outcomes of fresh IVF/ICSI cycles in populations of POSEIDON groups 1 and 2: a retrospective cohort study

Yuan Zhou[1,2], Yanying Lin[1,2], Lili Chen[1,2], Lisheng Huang[1,2], Yizhen Yang[1,2], Beihong Zheng[1,2] and Yan Sun[1,2]

[1] Center of Reproductive Medicine, Fujian Maternity and Child Health Hospital, Fujian Medical University, Fuzhou, China
[2] Fujian Maternal-Fetal Clinical Medicine Research Center, Fuzhou, China

Corresponding authors
Beihong Zheng,
zhengbeihong2010@163.com
Yan Sun, sunyan@fjsfy.com

## ABSTRACT

**Objective:** To investigate the impact of changes in luteinizing hormone (LH) levels during ovulation induction using flexible gonadotropin-releasing hormone antagonist (GnRH-A) regimens in POSEIDON groups 1 and 2 on the clinical outcomes of fresh *in-vitro* fertilization (IVF)/intracytoplasmic sperm injection (ICSI) cycles.

**Method:** A retrospective analysis was conducted on the clinical data of females in POSEIDON groups 1 and 2 who underwent IVF/ICSI using a flexible GnRH-A regimen at the Reproductive Medicine Center of Fujian Maternal and Child Health Hospital from January 2017 to December 2022. According to the values of change in LH [(LH level on the trigger day − baseline LH)/baseline LH] × 100%, the study subjects were divided into five groups. The general information, laboratory indicators, and clinical outcomes of each group were compared.

**Results:** In POSEIDON group 1, a significant difference was observed in terms of the number of retrieved eggs, mature eggs, fertilization number (two pronuclei (2PN) number), and cleavage among all groups (all $P$ values < 0.05). In POSEIDON group 2, a significant difference was observed in terms of the number of retrieved eggs, mature eggs, fertilization number (2PN number), cleavage, excellent embryos, and blastocysts (all $P$ values < 0.05). The number of eggs retrieved in group A was significantly lower than that in groups D and E ($P = 0.010$ and 0.001, respectively). The number of mature eggs in group A was significantly lower than that in groups D and E ($P = 0.008$ and 0.000, respectively). The number of mature eggs in group E was significantly higher than that in groups B and C ($P = 0.033$ and 0.021, respectively). The 2PN number in group A was significantly lower than that in groups D and E ($P = 0.042$ and 0.002, respectively). The cleavage count of group E was significantly higher than that of groups A and C ($P = 0.005$ and 0.037, respectively). The excellent embryos in group B were significantly lower than those of group E ($P = 0.038$).

**Conclusion:** The use of a flexible GnRH-A regimen to induce ovulation in the POSEIDON groups 1 and 2 can lead to better-controlled ovarian stimulation (COS) outcomes. This involves the LH level on trigger day decreasing within the range of

0–100% from baseline (with optimal range being ≥ −100% to −50%). This trend was more obvious in the POSEIDON group 2 population.

# INTRODUCTION

During the decades of flourishing advances in assisted reproductive technology, many ovulation induction regimens have been developed (*Zhou et al., 2024*). However, currently, the most commonly used and prevalent regimens in clinical practice are the gonadotropin-releasing hormone agonist (GnRH-a) and GnRH-A regimens. Compared with the GnRH-a regimen, the GnRH-A regimen exhibits advantages such as short treatment time, low gonadotropin (Gn) dosage, flexible medication, rapid inhibition of luteinizing hormone (LH) peak, and low incidence of ovarian hyper-stimulation syndrome (OHSS). It is the most commonly used controlled ovulation induction regimen for high ovarian response (HOR) and poor ovarian response (POR). Although experts from China have reached a consensus on the standardized application of GnRH-A regimens (*Expert Consensus Compilation Group of Reproductive Medicine Committee of China Medical Women's Association, 2022*), clinicians still need to continuously explore and improve the control and optimization of key nodes in the use of GnRH-A regimens in practice. In the process of antagonist therapy in COS, reasonably controlling the serum LH value is challenging. Early onset and sudden LH peaks can lead to premature luteinization of follicles, affecting the rate of egg retrieval and endometrial receptivity. However, LH itself is a substrate for estrogen synthesis and is essential in the middle and late stages of follicular development and final maturation. Excessive suppression of LH may lead to a high proportion of immature eggs and insufficient luteal function (*Wang et al., 2021*). Recent studies have reported that baseline LH and LH fluctuations during the antagonist COS process do not affect the clinical outcomes of patients. Even if LH temporarily increases, timely addition of antagonists and rapid decrease in LH levels will not have adverse effects on the clinical outcomes of *in-vitro* fertilization (IVF) (*Liu & Wang, 2023*; *Huang et al., 2023*; *Xiao et al., 2022*). However, previous studies have mostly focused on the HOR or entire population, and only few reports have focused on the regulation of antagonist regimens in the POR population. In clinical practice, it is difficult to precisely regulate the level of LH in individuals. Therefore, in this study, we focused on elucidating a more convenient observation method to monitor the changes in LH and optimize the clinical outcomes of patients by adjusting the drugs.

Previously, the Bologna criteria were often used to define POR. In 2016, the POSEIDON group shifted its focus from "low responsiveness" to "low prognosis" and proposed a POSEIDON stratification approach, dividing the POR population into four subgroups. Among them, groups 1 and 2 had normal ovarian reserve function (AFC ≥ 5 and AMH ≥ 1.2 ng/mL); however, unexpected POR or suboptimal response occurred (*Alviggi et al., 2016*). Regulating the COS regimen in this population can achieve the same live birth

rate as the population with normal ovarian response (*Parimala et al., 2021*). In this study, we retrospectively analyzed the population of POSEIDON groups 1 and 2 who underwent *in-vitro* fertilization (IVF)/intracytoplasmic sperm injection (ICSI) with flexible antagonist regimen. Furthermore, the correlation between changes in LH level during the COS process and clinical outcomes of fresh cycle were investigated. This study can help clinicians to assess the COS process with antagonist protocol, optimize the ovulation promotion plan, and improve clinical outcomes of patients.

# MATERIALS AND METHODS

## Research subjects and grouping

Retrospective analysis was performed using clinical data of POSEIDON groups 1 and 2 treated with the flexible IVF/ICSI regimen of GnRH-A at the Reproductive Medicine Center of Fujian Maternal and Child Health Hospital from January 2017 to December 2022. The inclusion criteria were as follows: (1) POSEIDON group 1 population [age < 35 years, normal ovarian reserve function (AFC ≥ 5 and AMH ≥ 1.2 ng/mL)] with unexpected ovarian low response or suboptimal response (number of retrieved eggs ≤ 9) or POSEIDON group 2 population [age ≥ 35 years and normal ovarian reserve function (AFC ≥ 5 and AMH ≥ 1.2 ng/mL)] with unexpected ovarian low response or suboptimal response (number of retrieved eggs ≤ 9); (2) flexible COS using antagonists. Exclusion criteria: (1) polycystic ovary syndrome; (2) multiple uterine fibroids (or single fibroid ≥ 5 cm); (3) history of intrauterine adhesions; (4) ovarian tumors; (5) uterine adenomyosis; (6) endometriosis uterine malformation; and (7) chromosomal abnormalities in either spouse causing recurrent implantation failure (RIF) or recurrent miscarriage (RSA).

According to the value of LH change [(LH level on the trigger day – baseline LH)/ baseline LH] × 100%, the study subjects were divided into five groups: groups A, B, C, D and E with the value of LH change ≥100%, ≥ 50% to <100%, ≥0% to <50%, ≥−50% to 0%, and ≥−100% to −50%, respectively.

This study was performed in accordance with the Declaration of Helsinki and was approved by the Medical Research Ethics Committee of Fujian Maternity and Child Health Hospital (Ethics approval number: 2024KY057). All patients are exempt from informed consent.

## Flexible GnRH-A regimens

Starting from the days 2–5 of menstruation, Gn (Recombinant Human Follitropin Alfa Solution for Injection; Merck, Germany; Procon, Merck, USA or urofollitropin for injection (Zhuhai Lizhu Pharmaceutical Company, Zhuhai, Guangdong, China)) was administered with an initial dose of 100–600 IU after considering the patient's age, body mass index (BMI), AMH, basal follicle stimulating hormone (FSH), and antral follicle count (AFC) levels. After Gn activation, serum FSH, LH, estradiol (E2), and progesterone (Prog) levels were checked every 3–5 days. Antagonist was added when the diameter of the dominant follicle reached 14 mm or approximately 15 mm or when the diameter of the dominant follicle was >12 mm and E2 > 300 ng/L (1 ng/L = 3.672 pmol/L). The dosage was adjusted till trigger day based on the follicle size and hormone levels. When the diameter of

three dominant follicles was ≥17 mm, that of two dominant follicles was ≥18 mm, or that of one dominant follicle was ≥20 mm, trigger was induced by administering recombinant human chorionic gonadotropin (hCG) (Aize, Merck, Switzerland) 250 μg or hCG (Zhuhai Lizhu Group Lizhu Pharmaceutical Factory, Guangdong, China) 6,000–10,000 IU. Egg retrieval was performed after 36 h.

## Laboratory indicators

Pronuclei were assessed 16–18 h post-insemination (IVF/ICSI), with 2PN indicating normal fertilization. Embryo culture was performed in a benchtop incubator (COOK Medical, Bloomington, IN, USA) under tightly controlled conditions. Cleavage-stage embryos were graded according to the Istanbul consensus, with Grade I and II embryos classified as high quality (*Alpha Scientists in Reproductive Medicine and ESHRE Special Interest Group of Embryology, 2011*). Blastocyst evaluation was performed using the Gardner scoring system, where blastocysts scoring ≥3BB (including AA, AB, BA, and BB) were considered high quality (*Gardner et al., 2000*).

The laboratory indicators included number of retrieved eggs, rate of egg retrieval, number of fertilization (2PN), fertilization rate, number of cleavage, cleavage rate, number of excellent embryos, rate of excellent embryos, number of blastocysts, and rate of blastocyst formation. The relevant indicators were assessed *via* referring to the consensus of experts on quality control of clinical key indicators of assisted reproductive technology, which included

egg retrieval rate = (the number of eggs retrieved/number of follicles penetrated) × 100%

fertilization rate (2PN fertilization rate) = (number of double pronuclear fertilizations/number of retrieved eggs) × 100%

cleavage rate = (number of cleavage/number of double pronuclear fertilization) × 100%

excellent embryo rate = (number of excellent embryos/number of cleavage) × 100%

blastocyst formation rate = (number of blastocysts/number of blastocyst cultures) × 100%.

## Clinical indicators

Clinical indicators included fresh cycle clinical pregnancy rate and live birth rate, which were calculated as follows:

clinical pregnancy rate (per fresh transplant cycle) = (number of clinical pregnancy cycles/number of fresh transplant cycles) × 100%

live birth rate (per fresh transfer cycle) = (number of live births/number of fresh embryo transfer cycles) × 100%.

## Statistical analysis

SPSS23.0 was used for statistical analysis. Quantitative data that conform to normal distribution were expressed as mean ± standard deviation. Further, homogeneity of variance testing was performed. Non-normally distributed data were represented by median and interquartile intervals. The Kruskal-Wallis H-test (KW test) was used for

intergroup comparisons, and Bonferroni method was used for pairwise comparisons between groups. Counting data were calculated using the Chi-square test. $P < 0.05$ was considered significant. $\chi^2$ test was used for counting data, which was expressed in terms of composition ratio.

## RESULT

A total of 733 patient data meeting the criteria were included. Among them, there were 408 cases in POSEIDON groups 1 (12 cases in group A, 13 cases in group B, 42 cases in group C, 109 cases in group D, 232 cases in group E), and 325 cases in POSEIDON groups 2 (13 cases in group A, 20 cases in group B, 42 cases in group C, 117 cases in group D, 133 cases in group E). There were 467 fresh transplantation cycles (269 cases in POSEIDON groups 1 and 198 cases in POSEIDON groups 2), 173 cases of clinical pregnancy (119 cases in POSEIDON groups 1 and 54 cases in POSEIDON groups 2), and 130 cases of live birth (95 cases in POSEIDON groups 1 and 35 cases in POSEIDON groups 2).

### Comparison of general information of patients in each group

In POSEIDON groups 1, no significant differences were observed in terms of the age, types of infertility, years of infertility, initial Gn dose, total Gn, and Prog level and endometrial thickness on the trigger day among the groups (all $P$ values $\geq 0.05$). The differences were significant among groups in terms of BMI, time to antagonist addition, Gn days, baseline LH, AMH, LH on the trigger day (all $P$ values $< 0.05$; Table 1a).

In POSEIDON groups 2, no significant differences were observed in terms of the age, BMI, types of infertility, years of infertility, total Gn, Gn days, AMH, and Prog level and endometrial thickness on the trigger day among the groups (all $P$ values $\geq 0.05$). The differences were significant among groups in terms of time to antagonist addition, initial Gn dose, baseline LH, LH on the trigger day (all $P$ values $< 0.05$; Table 1b).

### Comparison of laboratory indicators and clinical outcomes among various groups

In POSEIDON groups 1, no significant difference ($P \geq 0.05$) was observed among the groups in terms of egg retrieval rate, fertilization rate, cleavage rate, number of excellent embryos, excellent embryo rate, blastocyst number, and blastocyst formation rate. However, significant difference ($P$ value $< 0.05$) was observed in terms of number of retrieved eggs, mature egg number, fertilization number (2PN number), and number of cleavage ($P < 0.05$; Table 2a). Further grouping comparison indicated that in terms of number of double pronuclear fertilizations (2PN), only group E patients were significantly higher than group A patients ($P = 0.037$; Fig. 1).

In POSEIDON groups 2, no significant difference ($P \geq 0.05$) was observed among the groups in terms of egg retrieval rate, fertilization rate, cleavage rate, excellent embryo rate, and blastocyst formation rate. However, significant difference ($P$ value $< 0.05$) was observed in terms of number of retrieved eggs, mature egg number, fertilization number (2PN number), number of cleavage, number of excellent embryos, and blastocyst number ($P < 0.05$; Table 2b). In terms of the number of eggs retrieved, group A exhibited

**Table 1 Comparison of general information (POSEIDON groups 1 and 2).**

| | A ≥100% | B ≥50~<100% | C ≥0~<50% | D ≥−50~<0% | E ≥−100~<−50% | P value |
|---|---|---|---|---|---|---|
| a) Comparison of general information (POSEIDON groups 1) | | | | | | |
| No. of patients | 12 | 13 | 42 | 109 | 232 | |
| Age (years) | 29.17 ± 2.37 | 29.92 ± 3.28 | 29.74 ± 2.90 | 31.00 (5.00) | 30.50 (4.00) | 0.298 |
| BMI (kg/m$^2$) | 23.80 ± 2.57 | 20.20 ± 2.82 | 22.27 (3.04) | 22.18 (3.31) | 21.07 (3.97) | **0.002** |
| Primary infertility | 7/12 | 7/13 | 25/42 | 59/109 | 121/232 | 0.924 |
| Secondary infertility | 5/12 | 6/13 | 17/42 | 50/109 | 111/232 | |
| Duration of fertility (years) | 3.67 ± 2.53 | 3.00 (5.00) | 2.00 (4.00) | 3.00 (3.50) | 3.00 (3.00) | 0.193 |
| Time to antagonist addition (days) | 8.08 ± 2.87 | 5.85 ± 0.99 | 6.50 (1.00) | 7.00 (2.00) | 7.00 (2.00) | **0.044** |
| Initial Gn dose (IU) | 225.00 (62.50) | 225.00 (75.00) | 225.00 (25.00) | 225.00 (25.00) | 225.00 (75.00) | 0.181 |
| Total Gn (IU) | 2,545.83 ± 690.92 | 1,990.38 ± 619.27 | 2,100.00 (700.00) | 2,250.00 (937.50) | 2,025.00 (825.00) | 0.137 |
| Gn days | 10.83 ± 1.85 | 9.00 ± 1.41 | 10.00 (2.00) | 10.00 (2.00) | 10.00 (2.00) | **0.014** |
| Baseline LH (mIU/L) | 1.71 ± 0.75 | 1.92 ± 0.78 | 2.05 (1.72) | 2.80 (1.80) | 3.70 (1.91) | **0.000** |
| AMH (ng/ml) | 1.99 (0.97) | 1.75 (0.87) | 2.02 (2.59) | 2.28 (1.58) | 2.69 (2.58) | **0.007** |
| LH on the trigger day (mIU/L) | 4.86 ± 2.71 | 3.36 ± 1.43 | 2.30 (2.43) | 1.90 (1.05) | 1.00 (0.73) | **0.000** |
| Prog on the trigger day (ng/ml) | 0.55 ± 0.25 | 0.48 ± 0.21 | 0.70 ± 0.39 | 0.60 (0.50) | 0.60 (0.63) | 0.368 |
| Endometrial thickness on the Trigger day (mm) | 11.23 ± 2.38 | 10.89 ± 1.33 | 11.09 ± 1.66 | 11.00 (2.00) | 10.80 (2.50) | 0.448 |
| b) Comparison of general information (POSEIDON groups 2) | | | | | | |
| No. of patients | 13 | 20 | 42 | 117 | 133 | |
| Age (years) | 38.92 ± 2.84 | 38.00 (5.50) | 38.00 (4.25) | 39.00 (4.00) | 38.00 (3.00) | 0.124 |
| BMI (kg/m$^2$) | 22.82 ± 2.93 | 22.35 (3.79) | 22.79 ± 3.04 | 22.04 (3.72) | 21.86 ± 2.59 | 0.208 |
| Primary infertility | 4/13 | 3/20 | 9/42 | 28/117 | 30/133 | 0.862 |
| Secondary infertility | 9/13 | 17/20 | 33/42 | 89/117 | 103/133 | |
| Duration of fertility (years) | 2.00 (6.00) | 3.00 (2.75) | 3.00 (6.00) | 3.00 (3.00) | 4.00 (5.00) | 0.434 |
| Time to antagonist addition (days) | 7.23 ± 1.74 | 7.00 (2.75) | 7.00 (2.00) | 6.00 (2.00) | 7.00 (1.00) | **0.017** |
| Initial Gn dose (IU) | 225.00 (62.50) | 225.00 (75.00) | 225.00 (37.50) | 225.00 (12.50) | 225.00 (0.00) | **0.049** |
| Total Gn (IU) | 2,563.46 ± 859.91 | 2,808.75 ± 1,240.15 | 2,670.54 ± 801.12 | 2,250.00 (750.00) | 2,400 (750.00) | 0.212 |
| Gn days | 10.31 ± 2.39 | 9.95 ± 2.54 | 10.00 (2.25) | 10.00 (2.00) | 10.00 (2.50) | 0.085 |
| Baseline LH (mIU/L) | 2.20 (1.58) | 2.36 ± 1.29 | 2.41 ± 1.14 | 3.00 (1.74) | 3.60 (1.75) | **0.000** |
| AMH (ng/ml) | 1.53 (1.42) | 1.98 (0.77) | 1.72 (1.25) | 2.00 (1.55) | 2.11 (1.29) | 0.079 |
| LH on the trigger day (mIU/L) | 5.95 ± 3.52 | 3.96 ± 2.08 | 2.79 ± 1.38 | 2.20 (1.22) | 1.00 (0.60) | **0.000** |
| Prog on the trigger day (ng/ml) | 0.50 (1.09) | 0.80 ± 0.52 | 0.52 (0.44) | 0.60 (0.44) | 0.63 (0.68) | 0.474 |
| Endometrial thickness on the Trigger day (mm) | 11.17 ± 1.88 | 10.60 ± 2.24 | 10.87 ± 1.80 | 10.87 ± 1.81 | 10.80 (3.00) | 0.911 |

**Note:**
Quantitative data that conform to normal distribution were expressed as mean ± standard deviation. Non-normally distributed data were represented by median and interquartile range. BMI, body mass index; Gn, gonadotropin; LH, luteinizing hormone. *P* values written in bold indicate statistical significance.

significantly lower number than groups D and E (*P* = 0.010 and 0.0000, respectively; Fig. 2). In terms of the number of mature eggs, group A exhibited significantly lower number than groups D and G (*P* = 0.008 and 0.0000, respectively; Fig. 2), group B

**Table 2 Comparison of laboratory indicators (POSEIDON groups 1 and 2).**

| | A ≥100% | B ≥50~<100% | C ≥0%~<50% | D ≥−50%~<0% | E ≥−100%~<−50% | P value |
|---|---|---|---|---|---|---|
| **a) Comparison of laboratory indicators (POSEIDON group 1)** | | | | | | |
| Number of retrieved eggs | 4.58 ± 2.97 | 4.92 ± 2.63 | 6.00 (4.00) | 6.00 (3.00) | 7.00 (3.00) | **0.038** |
| Retrieved eggs rate (%) | 93.75 (47.50) | 100.00 (18.35) | 100.00 (21.25) | 100.00 (14.30) | 100.00 (20.00) | 0.288 |
| Number of mature eggs | 4.17 ± 2.79 | 4.77 ± 2.74 | 6.00 (4.00) | 5.00 (3.00) | 6.00 (3.75) | **0.007** |
| Number of double pronuclear fertilizations (2PN) | 3.08 ± 1.98 | 4.15 ± 2.34 | 5.00 (3.25) | 4.00 (3.00) | 5.00 (4.00) | **0.009** |
| Fertilization rate (%) | 72.82 ± 32.64 | 100.00 (22.5) | 84.50 (34.35) | 83.3 (40.00) | 85.70 (33.30) | 0.709 |
| Number of cleavage | 3.00 ± 1.86 | 3.77 ± 2.20 | 4.40 ± 2.30 | 4.00 (4.00) | 5.00 (3.75) | **0.024** |
| Cleavage rate (%) | 100.00 (0.00) | 100.00 (0.00) | 100.00 (0.00) | 100.00 (0.00) | 100.00 (0.00) | 0.918 |
| Number of excellent embryos | 1.92 ± 1.44 | 2.92 ± 1.80 | 3.00 (3.25) | 3.00 (3.00) | 3.00 (3.00) | 0.541 |
| Excellent embryos rate (%) | 59.58 ± 35.13 | 100.00 (46.43) | 58.57 (60.63) | 66.67 (31.67) | 66.67 (42.50) | 0.211 |
| Number of blastocysts | 0.00 (1.00) | 0.00 (1.00) | 0.50 (2.00) | 1.00 (2.00) | 1.00 (2.00) | 0.597 |
| Blastocyst formation rate (%) | 0.00 (100.00) | 0.00 (75.00) | 12.50 (85.00) | 33.33 (75.00) | 25.00 (82.50) | 0.955 |
| **b) Comparison of laboratory indicators (POSEIDON groups 2)** | | | | | | |
| Number of retrieved eggs | 3.08 ± 2.56 | 4.65 ± 2.58 | 5.19 ± 2.21 | 5.00 (4.00) | 6.00 (3.00) | **0.00** |
| Retrieved eggs rate (%) | 88.90 (45.00) | 100.00 (15.00) | 100.00 (28.60) | 10.00 (20.00) | 100.00 (18.35) | 0.368 |
| Number of mature eggs | 2.77 ± 2.24 | 4.20 ± 2.21 | 4.5 (3.00) | 5.00 (4.00) | 6.00 (4.00) | **0.00** |
| Number of double pronuclear fertilizations (2PN) | 2.62 ± 2.29 | 4.00 (2.75) | 4.00 (2.50) | 4.00 (3.00) | 5.00 (3.00) | **0.00** |
| Fertilization rate (%) | 100.00 (50.00) | 100.00 (31.22) | 87.50 (38.13) | 87.50 (33.30) | 100.00 (22.20) | 0.516 |
| Number of cleavage | 2.54 ± 2.30 | 3.55 ± 2.46 | 3.00 (3.25) | 4.00 (3.00) | 5.00 (4.00) | **0.000** |
| Cleavage rate (%) | 100.00 (16.65) | 100.00 (10.72) | 100.00 (25.00) | 100.00 (0.00) | 100.00 (0.00) | 0.634 |
| Number of excellent embryos | 1.92 ± 1.89 | 2.00 (2.75) | 2.00 (2.00) | 3.00 (3.00) | 3.00 (2.50) | **0.003** |
| Excellent embryos rate (%) | 62.73 ± 83.33 | 37.50 (66.67) | 60.79 ± 32.20 | 66.67 (35.71) | 66.67 (33.33) | 0.094 |
| Number of blastocysts | 0.00 (0.50) | 0.00 (1.00) | 0.00 (1.25) | 0.00 (2.00) | 1.00 (2.00) | **0.045** |
| Blastocyst formation rate (%) | 0.00 (30.00) | 0.00 (50.00) | 0.00 (83.33) | 0.00 (66.67) | 50.00 (77.50) | 0.112 |

**Note:**
Quantitative data that conform to normal distribution were expressed as mean ± standard deviation. *P* values written in bold indicate statistical significance.

exhibited significantly lower number than group E (*P* = 0.033; Fig. 2) and group C exhibited significantly lower number than group E (*P* = 0.021; Fig. 2). In terms of fertilization number (2PN number), group A exhibited significantly lower number than groups D and E (*P* = 0.042 and 0.002, respectively; Fig. 2). In terms of cleavage count, group E exhibited significantly higher value than groups A and C (*P* = 0.005 and 0.037, respectively; Fig. 2). In terms of excellent embryos, group B exhibited significantly lower number than groups E (*P* = 0.038; Fig. 2).

No significant difference was observed in terms of clinical pregnancy rate and live birth rate among the groups (all *P* ≥ 0.05; Tables 3a and 3b).

## DISCUSSION

Follicular development is governed by a physiological mechanism known as the "two-cell, two-gonadotrophin" model. In this process, LH binds to receptors on theca cells, stimulating androgen production. These androgens are then transported to granulosa cells,

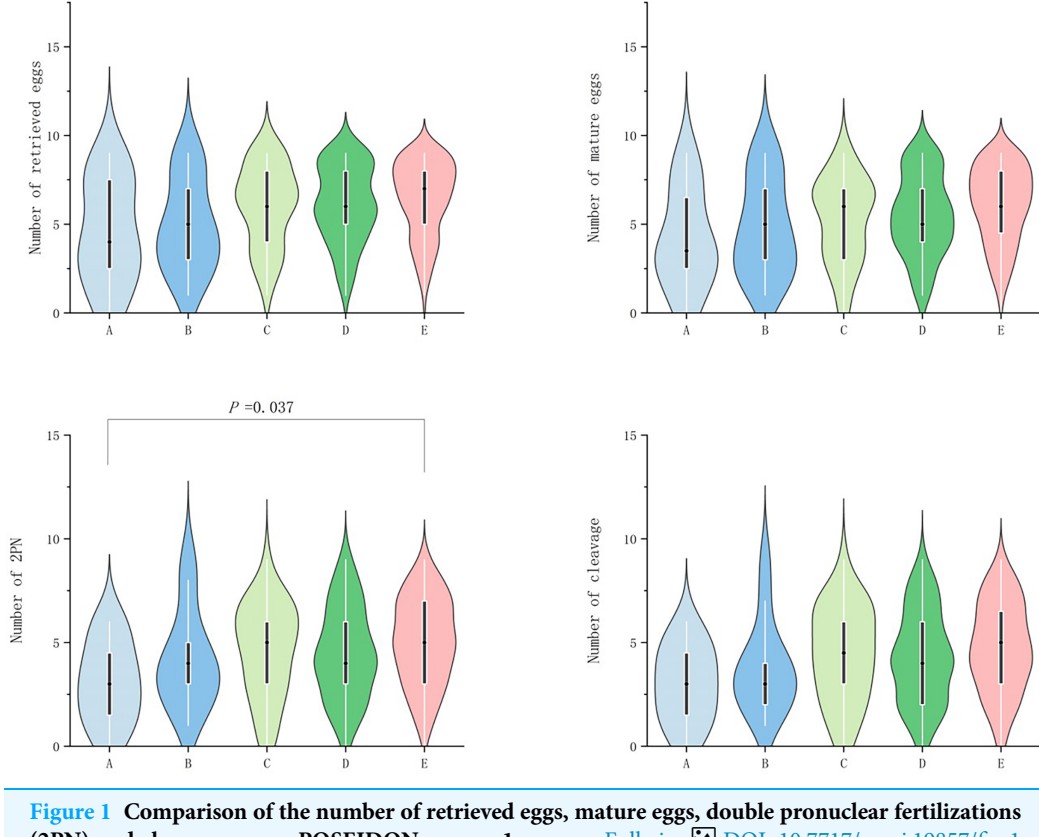

**Figure 1 Comparison of the number of retrieved eggs, mature eggs, double pronuclear fertilizations (2PN) and cleavage across POSEIDON groups 1.**

where they are converted into estradiol *via* aromatase activity. When a follicular wave with a low FSH threshold becomes dominant, LH levels surge sharply. Notably, the LH threshold for dominant follicles is significantly higher than that of non-dominant follicles. This disparity halts mitosis in granulosa cells of non-dominant follicles, leading to their atresia, while the dominant follicle progresses to maturity. Thus, elevated LH levels suppress further growth of non-dominant follicles—a phenomenon termed the "LH ceiling" hypothesis (*Kumar & Sait, 2011*). Similarly, previous studies have proposed the concept of "LH treatment window," believing that LH levels above or below the threshold level will lead to insufficient E2 levels, thereby affecting follicular development (*Shoham, 2002*). Some studies suggested that the rate and direction of LH changes determine follicular development, and basline LH values and LH fluctuations before the addition of antagonists do not affect patient outcomes (*Liu & Wang, 2023*; *Orvieto et al., 2021*; *Huirne et al., 2005*; *Zhou et al., 2023*). However, *Zhang et al. (2019)* believed that although the application of antagonists can inhibit LH surge, it will have adverse effects on embryonic development and pregnancy outcomes in fresh transfer cycles. Therefore, the control of LH levels during the process of ovulation promotion is debatable. The key to the flexible use of antagonists lies in how to regulate LH levels during the COS process through timely and appropriate addition of antagonists, thereby improving patient egg retrieval, embryo retrieval, and clinical outcomes (*Expert Consensus Compilation Group of Reproductive*

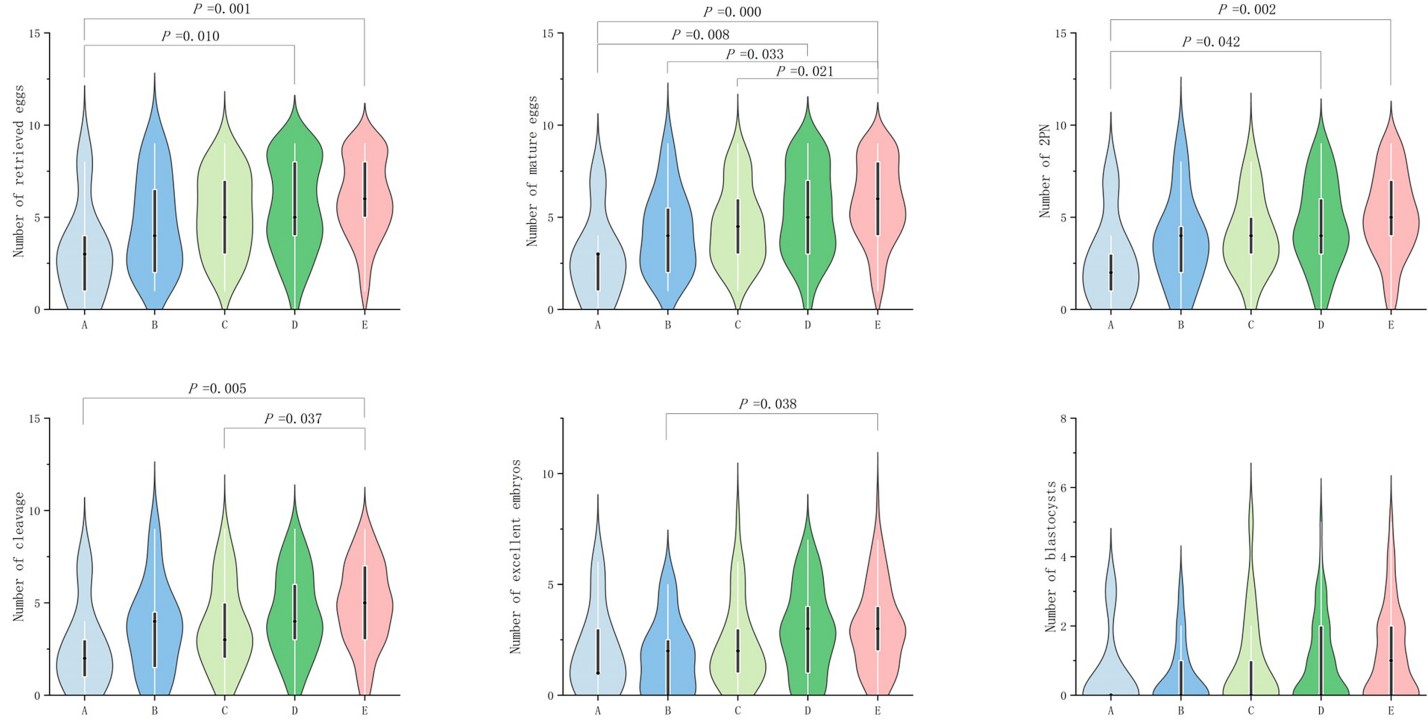

**Figure 2 Comparison of the number of retrieved eggs, mature eggs, double pronuclear fertilizations (2PN), cleavage, excellent embryos and blastocysts across POSEIDON groups 2.**

**Table 3 Comparison of clinical outcomes (POSEIDON groups 1 and 2).**

| | A ≥100% | B ≥50~<100% | C ≥0~<50% | D ≥−50~<0% | E ≥−100~<−50% | $\chi^2$ | P value |
|---|---|---|---|---|---|---|---|
| a) Comparison of clinical outcomes (POSEIDON groups 1) | | | | | | | |
| Clinical pregnancy rate (%) | 57.14 (4/7) | 33.3 (4/12) | 37.04 (10/27) | 46.58 (34/73) | 44.67 (67/150) | 1.791 | 0.774 |
| Live birth rate (%) | 28.57 (2/7) | 33.3 (4/12) | 33.33 (9/27) | 38.36 (28/73) | 34.67 (52/150) | 0.561 | 0.979 |
| b) Comparison of clinical outcomes (POSEIDON groups 2) | | | | | | | |
| Clinical pregnancy rate (%) | 0.00 (0/6) | 23.08 (3/13) | 33.33 (9/27) | 25.35 (18/71) | 29.63 (24/81) | 2.867 | 0.597 |
| Live birth rate (%) | 0.00 (0/6) | 15.38 (2/13) | 25.93 (7/27) | 15.49 (11/71) | 18.52 (15/81) | 2.328 | 0.671 |

Notes:
Clinical pregnancy rate = (number of clinical pregnancy cycles/number of fresh transplant cycles) × 100%.
Live birth rate = (number of live births/number of fresh embryo transfer cycles) × 100%.

*Medicine Committee of China Medical Women's Association, 2022*). In clinical practice, medication dosage is generally adjusted every few days based on the patient's follicular development and hormonal changes. It is difficult to accurately control the LH changes at each time node, and the global view of LH fluctuations can only be reflected on the trigger day. If unexpected LH fluctuations occur, timely remedies may not be possible. Therefore, it is necessary to explore a more convenient and intuitive way to describe LH fluctuations and guide clinicians to regulate serum LH values based on outcomes. In this study, the patients were divided based on overall changes in LH during the COS process [(LH level on

the trigger day − baseline LH)/baseline LH] × 100%, and the laboratory and clinical outcomes of each group were compared. Compared with the patients with an increase in LH levels, those with a decrease in LH levels exhibited more number of retrieved eggs, mature eggs, 2PN, cleavage, excellent embryos, and blastocysts. Among them, group E exhibited superior performance (LH change ≥−100% to −50%) and this trend was more obvious in the POSEIDON groups 2 population. This is similar to the research findings of *Wang et al. (2017)*, *Ji et al. (2017)*, who believed that LH levels on trigger day are negatively correlated with the number of retrieved eggs but have no significant impact on the clinical pregnancy outcomes of patients. This suggested that during the COS process, maintaining the LH level within the range of 0–100% lower than the baseline (with ≥−100% to −50% being optimal) can lead to better egg and embryo outcomes.

Patients with low ovarian response have poorer pregnancy outcomes *via* assisted reproductive technology (ART) and higher treatment costs compared with those with normal ovarian response. However, if the COS regimen can be reasonably grasped, they can achieve the same clinical outcomes as the normal population (*Chinese Association of Reproductive Medicine, 2022*). This study positioned the target population as the POR under the POSEIDON classification. Since the POSEIDON classification is prognostic and emphasizes individualized ovarian stimulation, it has greater clinical value. Among the populations of groups 1 and 2 of POSEIDON, their ovarian reserve function is still acceptable but they exhibit unexpected low responses resulting in suboptimal egg retrieval. Previous studies have suggested that this may be related to mutations in Gn receptor genes or single nucleotide polymorphisms (SNPs) (*Alviggi et al., 2018*; *König et al., 2019*). Some scholars also believe that this may be related to the low initial dose of Gn (*Parimala et al., 2021*; *Drakopoulos et al., 2018*). Current evidence suggests that increasing FSH dosage with adjunctive LH supplementation may benefit certain patient populations (*Sunkara, Ramaraju & Kamath, 2020*; *Wang et al., 2023*). However, robust clinical data remain insufficient regarding optimal dosing regimens, FSH/LH ratios, and standardized efficacy assessment protocols. Our study focused on older patients with unexpected poor ovarian response (POSEIDON group 2), revealing an intriguing pattern: when trigger-day LH levels decreased by ≥100% to <50% from baseline values, we observed improved laboratory parameters, including oocyte yield. Notably, these biochemical improvements did not translate into significant differences in clinical outcomes—a finding consistent with existing literature (*Zhou et al., 2023*).

## Limitations

This study has several limitations. LH is secreted in a pulsatile manner, and a single measurement may not accurately reflect its dynamic fluctuations. Although we assessed the trend of LH changes using the formula [(LH level on the trigger day − baseline LH)/baseline LH] × 100%, the analysis still relies on discrete time-point measurements.

Additionally, most laboratory parameters were continuous variables, and some subgroups categorized by LH changes had relatively small sample sizes. The data distribution across groups did not fully meet the assumptions of normality, necessitating non-parametric statistical tests. While we mitigated potential confounding effects through

methods such as age stratification, some bias in the conclusions may persist. Future studies should aim to expand the sample size and employ more robust methodologies to further investigate the optimal control of key nodes in the flexible antagonist protocol.

## CONCLUSIONS

In summary, our study suggested that using a flexible antagonist regimen to induce ovulation in the POSEIDON groups 1 and 2 can lead to better COS outcomes by the LH level decreasing within the range of 0–100% from baseline (with ≥−100% to −50% being optimal), especially in POSEIDON groups 2.

## ACKNOWLEDGEMENTS

We would like to express our thanks to Bullet Edits Limited for their assistance in the linguistic editing and proofreading of this manuscript.

### Funding

This study was supported by grants from the Natural Science Foundation of Fujian Province (No. 2023J011229), Major Scientific Research Program for Young and Middle-aged Health Professionals of Fujian Province, China (grant No. 2022ZQNZD010), Innovation Platform Project of Science and Technology, Fujian province (2021Y2012) and Key Project on the Integration of Industry, Education and Research Collaborative Innovation of Fujian Province (grant No. 2021YZ034011). The funders had no role in study design, data collection and analysis, decision to publish, or preparation of the manuscript.

### Grant Disclosures

The following grant information was disclosed by the authors:
Natural Science Foundation of Fujian Province: 2023J011229.
Major Scientific Research Program for Young and Middle-aged Health Professionals of Fujian Province, China: 2022ZQNZD010.
Innovation Platform Project of Science and Technology, Fujian province: 2021Y2012.
Key Project on the Integration of Industry, Education and Research Collaborative Innovation of Fujian Province: 2021YZ034011.

### Competing Interests

The authors declare that they have no competing interests.

### Author Contributions

- Yuan Zhou conceived and designed the experiments, performed the experiments, analyzed the data, prepared figures and/or tables, and approved the final draft.
- Yanying Lin performed the experiments, prepared figures and/or tables, and approved the final draft.

- Lili Chen performed the experiments, prepared figures and/or tables, and approved the final draft.
- Lisheng Huang performed the experiments, prepared figures and/or tables, and approved the final draft.
- Yizhen Yang performed the experiments, prepared figures and/or tables, and approved the final draft.
- Beihong Zheng conceived and designed the experiments, authored or reviewed drafts of the article, and approved the final draft.
- Yan Sun conceived and designed the experiments, authored or reviewed drafts of the article, and approved the final draft.

## Human Ethics

The following information was supplied relating to ethical approvals (*i.e.*, approving body and any reference numbers):

This study was performed in accordance with the Declaration of Helsinki and was approved by the Medical Research Ethics Committee of Fujian Maternity and Child Health Hospital (Ethics approval number: 2024KY057).

## Data Availability

Raw data is available in the Supplemental Files.

## Supplemental Information

Supplemental information for this article can be found online at http://dx.doi.org/10.7717/peerj.19857#supplemental-information.

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
