# Peer review of "The impact of changes in LH levels in flexible antagonist protocols on the clinical outcomes of fresh IVF/ICSI cycles in populations of POSEIDON groups 1 and 2: a retrospective cohort study"

_PeerJ, doi:10.7717/peerj.19857_

## Round 0.1 · original submission · Major Revisions

·

Basic reporting

The Title and Abstract fully reveal what is discussed in the following material.
The introduction very well emphasizes the subtleties of GnRh treatments and the importance and sensitivity of LH in oocyte maturation.

Experimental design

Material and method
The study groups, Poseidon group, inclusion criteria, and exclusion criteria are described.
Line 104,105,106,107, where is this formula with LH from??
Line 119,120, the dosage was adjusted depending on the size of the follicles, and hormonal values - can you explain in more detail?

Validity of the findings

Explain what superior embryos mean??? Should international embryo grading systems be used? Istanbul Consensus, Gardner grading
How were the embryos cultured?? Culture media, incubators, classic incubator, time-lapse incubator, etc.
Based on two LH determinations, can we make an opinion?? Maybe serial LH determinations should be done?
The tables are too large with a lot of data, hard to follow, maybe they should be re-divided, hormonal values ​​in a separate table, clinical data in another table, considering that there are 7 batches.
I didn't understand the figures from 1-9.

Additional comments

From my point of view, reading an article should be a pleasure, even for laymen, to understand the flow and conclusions of the study.

I suggest a more precise explanation of the phenomena involved in follicle maturation, perhaps molecular explanations of the action of hormones.

Reviewer 2 ·

Basic reporting

The authors have submitted a well-written manuscript with clear and professional English throughout. The literature references are appropriate, providing sufficient background. However, the descriptions and explanations of the figures and tables are lacking, making the overall reading experience difficult. This aspect needs to be improved.

Experimental design

The study is retrospective in design, and an appropriate method of data analysis was selected. However, the rationale for dividing the study population into seven groups is unclear. This approach raises concerns that the authors may have been attempting to validate a preconceived hypothesis by testing multiple subgroup analyses until achieving the desired results.

If changes in LH levels are truly an independent factor influencing COS outcomes, a more robust statistical approach, such as regression analysis, should have been applied to confirm the independent effect of LH changes while controlling for potential confounding variables.

Validity of the findings

The manuscript is clearly written and well-structured, including all essential sections (Abstract, Introduction, Methods, Results, Discussion, Conclusions, References). The authors aimed to investigate the impact of changes in LH levels during COS on IVF outcomes. They categorized the study population into seven groups based on the change in LH levels on the trigger day compared to day 2, and concluded that a decrease of LH by 100% to 50% yields the best ovarian stimulation outcomes in women classified under Poseidon groups 1 and 2.

However, there are critical concerns that limit the validity of the findings:

The substantial differences in sample size across the groups, as well as variations in mean BMI and basal LH levels, could act as significant confounding factors influencing the COS outcomes.

The authors failed to adequately address or adjust for these confounders in their analysis, thereby compromising the reliability and clinical relevance of the study's conclusions.

Given these limitations, I am not in favor of accepting this manuscript for publication.

---

## Round 0.2 · Minor Revisions

Thank you for the revised version of your manuscript and for the careful attention given to the reviewers' comments. We appreciate the improvements made and your effort to enhance the robustness of the study.

Before we can proceed further, we kindly request that you include a dedicated "Limitations" section in your manuscript. This section should clearly present and discuss all potential sources of bias—both methodological and non-methodological—that may affect your study’s findings or interpretations. A transparent discussion of limitations is crucial to ensure the scientific integrity and reliability of your contribution.

We look forward to receiving your revised version.

·

Basic reporting

-

Experimental design

-

Validity of the findings

-

Additional comments

The comment is based on the laboratory tests to identify the LH parameter. As I know, LH is secreted in peaks. It is possible that a single determination is not specific to the patient. I would try in future tests performed over a certain time, ie, LH series (was a recommendation for future research)

---

## Round 0.3 · accepted · Accept

Thank you for submitting the revised version of your manuscript. After carefully reviewing the changes, I can confirm that all reviewers' comments and suggestions have been appropriately addressed.

Based on this assessment, I am pleased to inform you that the manuscript is now ready for publication. Congratulations.